# Soluble organic matter Molecular atlas of Ryugu reveals cold hydrothermalism on C-type asteroid parent body

Philippe Schmitt-Kopplin [1,2,3] ✉, Norbert Hertkorn [2], Mourad Harir[2], Franco Moritz[2], Marianna Lucio[2], Lydie Bonal[4], Eric Quirico [4], Yoshinori Takano [5], Jason P. Dworkin [6], Hiroshi Naraoka [7], Shogo Tachibana [8,9], Tomoki Nakamura[10], Takaaki Noguchi [11], Ryuji Okazaki[7], Hikaru Yabuta [12], Hisayoshi Yurimoto [13], Kanako Sakamoto [9], Toru Yada [9], Masahiro Nishimura[9], Aiko Nakato [9], Akiko Miyazaki [9], Kasumi Yogata [9], Masanao Abe[9], Tomohiro Usui [9], Makoto Yoshikawa[9], Takanao Saiki[9], Satoshi Tanaka[9], Fuyuto Terui[9], Satoru Nakazawa [9], Tatsuaki Okada [9], Sei-ichiro Watanabe [14], Yuichi Tsuda[9] & Hayabusa2-initial-analysis SOM team*

The sample from the near-Earth carbonaceous asteroid (162173) Ryugu is analyzed in the context of carbonaceous meteorites soluble organic matter. The analysis of soluble molecules of samples collected by the Hayabusa2 spacecraft shines light on an extremely high molecular diversity on the C-type asteroid. Sequential solvent extracts of increasing polarity of Ryugu samples are analyzed using mass spectrometry with complementary ionization methods and structural information confirmed by nuclear magnetic resonance spectroscopy. Here we show a continuum in the molecular size and polarity, and no organomagnesium molecules are detected, reflecting a low temperature and water-rich environment on the parent body approving earlier mineralogical and chemical data. High abundance of sulfidic and nitrogen rich compounds as well as high abundance of ammonium ions confirm the water processing. Polycyclic aromatic hydrocarbons are also detected in a structural continuum of carbon saturations and oxidations, implying multiple origins of the observed organic complexity, thus involving generic processes such as earlier carbonization and serpentinization with successive low temperature aqueous alteration.

[1]Technische Universität München, Analytische Lebensmittel Chemie, Maximus-von-Forum 2, 85354 Freising, Germany. [2]Helmholtz Munich, Analytical Bio-GeoChemistry, Ingolstaedter Landstraße 1, 85764 Neuherberg, Germany. [3]Max Planck Institute for Extraterrestrial Physics, Gießebachstraße 1, 85748 Garching bei München, Germany. [4]Université Grenoble Alpes, CNRS, CNES, IPAG, 38000 Grenoble, France. [5]Biogeochemistry Research Center (BGC), Japan Agency for Marine-Earth Science and Technology (JAMSTEC), 2-15 Natsushima, Yokosuka 237-0061, Japan. [6]Solar System Exploration Division, NASA Goddard Space Flight Center, Greenbelt, Maryland 20771, USA. [7]Department of Earth and Planetary Sciences, Kyushu University, Motooka 744, Nishiku, Fukuoka 819-0395, Japan. [8]Tokyo Organization for Planetary and Space Science, University of Tokyo, Bunkyo-ku, Tokyo 113-0033, Japan. [9]Institute of Space and Astronautical Science, Japan Aerospace Exploration Agency (ISAS/JAXA), Sagamihara 252-5210, Japan. [10]Department of Earth Material Science, Tohoku University, Aoba-ku, Sendai 980-8578, Japan. [11]Division of Earth and Planetary Sciences, Kyoto University, Kyoto 606-8502, Japan. [12]Department of Earth and Planetary Sciences, Hiroshima University, Higashi-Hiroshima, Hiroshima 739-8526, Japan. [13]Department of Earth and Planetary Sciences, Hokkaido University, Kita-ku, Sapporo 060-0810, Japan. [14]Graduate School of Environment Studies, Nagoya University, Nagoya 464-8601, Japan. *A list of authors and their affiliations appears at the end of the paper. ✉e-mail: schmitt-kopplin@tum.de

The soluble organic matter (SOM) in carbonaceous chondrites is chemically highly diverse and integrates in its chemistry much of the history on temperature events and water-rock interaction on the parent body[1–3]. Two touch-down samplings by the Hayabusa2 spacecraft in February and July 2019 enabled the collection of surface (samples stored in Chamber A) and possibly sub-surface materials (samples stored in Chamber C) of the near-Earth $C_b$-Type asteroid (162173) Ryugu. Samples were returned to Earth on December 6, 2020 and with about 5.4 g material that constitutes the fourth returned sample from extraterrestrial bodies following Apollo, Luna and recently Chang'e 5 from the Moon, Stardust from comet 81P/Wild2 and Hayabusa from the near-Earth S-type asteroid Itokawa, respectively. These material from enabled first time the comparison of our knowledge on meteorites to a a C-type asteroid[4,5]. Ryugu is considered as a primitive small Solar-System body and its low-albedo reflectance spectrum in the wavelength range of 1.8 to 3.0 μm[6] and spectroscopic similarity[7] made it from observation a good candidate in being a parent body of carbonaceous chondrites (CCs), rich in water and organics.

The analysis of the Ryugu material showed the dominance of hydrous silicate minerals including serpentine and saponite associated with dolomite, pyrrhotite and magnetite, reflecting episode of aqueous alteration[8] and suggesting similarities with CI chondrites[4,8,9]. Element analysis confirmed Ryugu with the richest C, N, and H concentrations compared to various types of CCs with signatures comparable to the observed falls Ivuna and Orgueil CI chondrites[4,10]. The analyzed elements content of C, H, N, S, and pyrolyzable O (not including O of anhydrous silica) was found to be ~21.3 wt% including the carbonates, the sulfides as well as the soluble organic matter (SOM) and the macromolecular insoluble organic matter (IOM)[5]. Targeted organic analysis of Ryugu A0106 sample revealed ~15 amino acids, including many non-protein amino acids in an approximately racemic mixture indicating a non-biological origin. Their abundances were different to Orgueil CI, reflecting different chemosynthetic or alteration pathway conditions on the asteroid parent bodies[4]. High abundances of β-alanine, γ-amino-n-butyric acid, and δ-aminovaleric acid in the Ryugu samples indicated a higher peak temperature than the CI chondrites as these compounds are typical in CV and CO chondrites processed at elevated temperatures of up to ~300 °C[5,11,12]. Series of aliphatic amines (i.e., methylamine ($CH_3NH_2$), ethylamine ($C_2H_5NH_2$) and isopropylamine, n-propylamine ($C_3H_7NH_2$)) and short chained carboxylic acids were analyzed in Ryugu samples as well[5]. Further targets were polycyclic aromatic hydrocarbons (PAH) as they are ubiquitous in presolar synthesis in interstellar locations[13] and the FTIR spectra of Ryugu samples were close to interstellar PAHs spectra, suggesting incorporation of interstellar PAHs to Ryugu during its accretion[5]. Four-ringed fluoranthene, pyrene, chrysene/triphenylene and methylated fluoranthene and pyrene as well as smaller PAHs as naphthalene, phenanthrene and anthracene were analyzed and their relative abundances were attributed to differential alteration survivals and solubilities in parent body fluids[5]. Alkylpyridines and alkylimidazoles (aromatic N-heterocycles) and alkylpiperidines (aliphatic N-heterocycles)[5,14] were targeted in Ryugu samples and showed a different profile as found earlier in CM2 Murchison[15,16] reflecting a different redox condition on the parent body.

The returned surface sample of Ryugu A0106 enable now a direct analysis of SOM from the surface of the parent body and the comparison with meteoritic SOM. Here, we report the molecular characteristics of Ryugu's SOM from non-targeted organic analysis. More specifically we compared the ultrahigh-resolution mass spectroscopy analysis profile of the methanol extracts to 36 CCs that experienced moderate temperatures and partial aqueous alteration to position Ryugu's organic diversity in the context of possible hydrothermalism.

## Results and discussion
### The molecular atlas of Ryugu SOM
The Ryugu A0106 sample was extracted sequentially with solvents of increasing polarity, starting with hexane, dichloromethane, methanol and water to generate/isolate chemical fractions of SOM with molecules of increasing polarity. We first optimized injection flow rates to generate maximum scan count and signal-to-noise response to fully use the minimum amount of 5 mg of available extracts for a maximal information output in mass spectrometry and NMR-spectroscopy. We used electrospray ionization in negative (ESI(-)) and positive (ESI(+)) modes as well as atmospheric pressure photoionization in positive mode (APPI(+)) in direct injection Fourier transform-ion cyclotron resonance mass spectrometry (DI-FT-ICR/MS) as previously tested and demonstrated powerful method on many meteorites earlier[3,17]. We also confirmed that the methanol extract of the sequential extraction procedure and the direct extraction of Murchison CM2 and Aguas Zarcas CM2 fragments converged in the same signal and compositional profiles, making these two methanol extracts directly comparable. One third of the small solution volume available was used for nuclear magnetic resonance (NMR) spectroscopy experiments.

All analyzed FTICR-MS spectra showed very high signal density in the mass range from m/z 120 to 700 very comparable when analyzing CCs, reflecting that rapidly collected CCs are not biased by terrestrial contamination and weathering. The methanol extract in ESI(-) showed regular intense signals of increased intensity that could be assigned to polythiols (disulfate) with a number of sulfur from S3 to S9 (Fig. S1). The less polar hexane and dichloromethane extracts further showed the presence of polythiols as monosulfonates and of polysulfanes with sulfur numbers S3–S8. From all CCs analyzed in this study (Table S1), such intense profiles were observed only in methanol extract of recently fallen Tarda C2-ung material[18]. Each nominal mass always showed multiple signals with a repeating pattern of mass spacing of 36.3845 mDa following a compositional substitution of $CH_4$ with respect to O reflecting overall structural regularity in chemical homologous series in the mixture[1]. Moreover, the Gaussian distribution of the mass signals within each nominal mass represents the large number of possible isomers for each exact mass, changing from a higher oxygen degree and carbon unsaturation from lower to higher mass defect as reported[19], making any detailed structural annotation nearly impossible. In ESI(+) and APPI(+) the same regularity pattern was observed as well, similarly to patterns earlier described with all meteorite SOMs[3,17].

In total, with all ionization modes and extractions we distinguished a molecular richness with more than 200,000 signals at S/N = 3. We could reduce them with conservative filter rules to 23,100 elementary compositions in the CHNOS monoisotopic atomic space. These results agree with our previous studies for various classes of CCs from a small molecule mass spectrometry perspective[2,3], demonstrating their huge chemical diversity compared to the biological space[20,21]. Figure 1A–C shows the final counting in elementary compositions assigned from FTICR-MS exact mass data and the overlaps of the compositional space between the different ionization methods and extraction solvents. The overall most abundant chemical families were the N-containing molecules (16,950 formula), followed by CHO/Na (8843 formula), CHOS/Na (5647 formula), CH (1056 formula) and CHS (165 formula). Most abundant molecules were extracted in polar protic methanol, followed by dichloromethane. Highly polar and highly apolar compounds extracted with water and hexane, respectively, showed the most specific compounds in terms of carbon oxidation state, nitrogen or sulfur content (Figs. S2, S4, and S5). In APPI(+) the oxygenation degree of CHO, CHNO and CHOS was similar and independent of the solvent polarity (2-3,O-atoms per molecule). The number of nitrogen, however, increased with increasing polarity indicating the involvement of nitrogen in polar functional groups such as primary or secondary amines increasing molecular basicity and thus

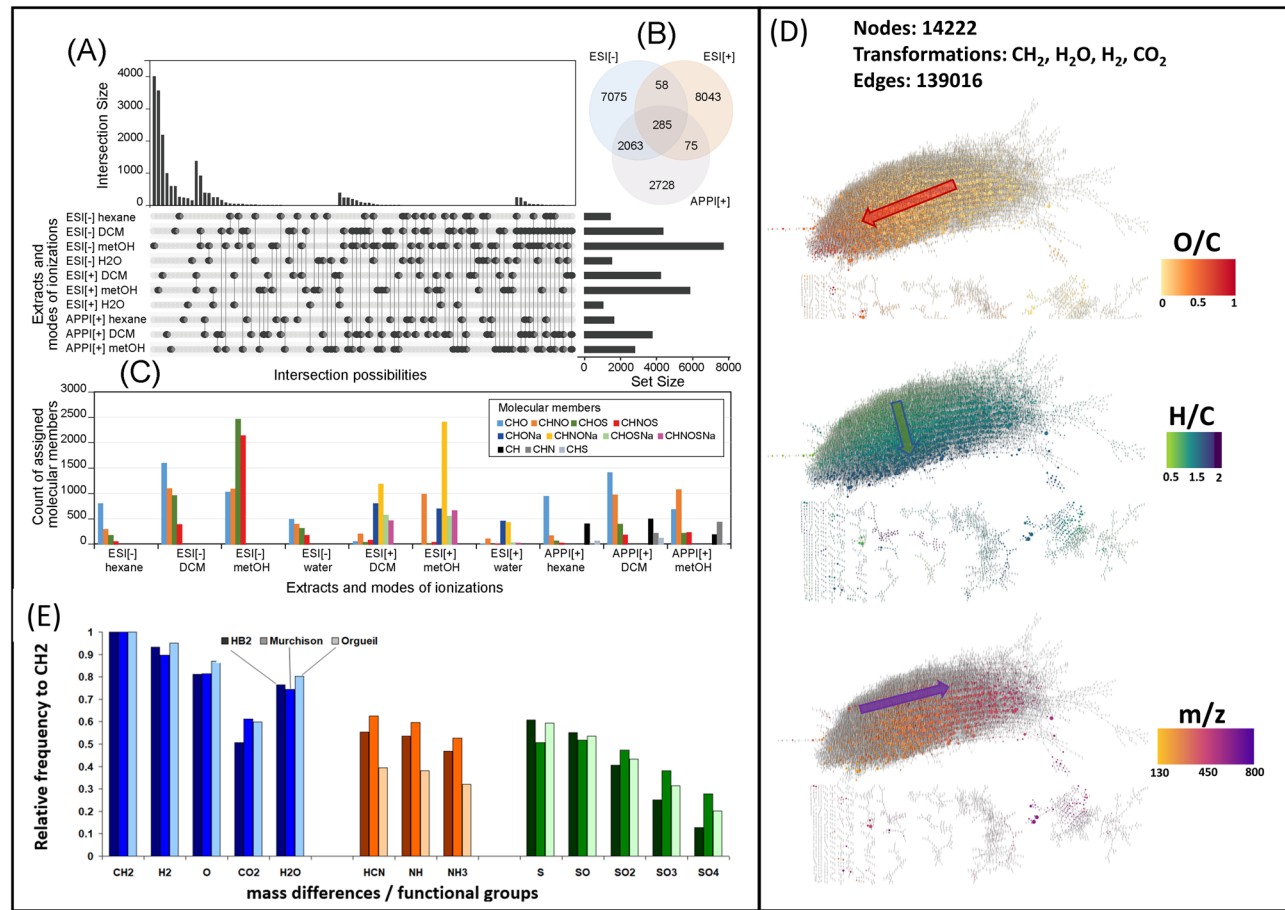

**Fig. 1 | Molecular atlas of the Ryugu soluble organic matter. A** counting of the number of elementary composition retrieved from exact mass analysis in ESI(-), ESI(+) and APPI(+) ionization modes of the sequential extractions in hexane, dichloromethane, methanol and water. **B** Venn-diagram of the ionization mode signatures, **C** abundances of the various chemical families in the various extracts and ionization modes, involving, CH, CHO, CHN, CHS, CHOS, CHNO, CHNOS, CHONa, CHNONa, CHOSNa and CHNOSNa. **D** Network representation of the methanol extract using the most frequent mass transitions and showing the strong structural connectivity and the regular gradients in O/C, H/C and $m/z$. **E** frequency histogram of the exact mass differences (i.e., $CH_2$, $H_2$, O, $CO_2$, $H_2O$, HCN, NH, $NH_3$, SO, S, $SO_2$, $SO_3$, and $SO_4$).

promoting APPI(+) ionization (Fig. S3). CH type of molecules detected in APPI(+) showed a continuum in abundance of aliphatic to highly aromatic compounds bearing multiple aromatic rings (Fig. S6). Similarly, in ESI(+), the oxygenation degree is not much affected in the various polarity extracts, but was higher than in APPI (3-4 oxygen atoms per molecule) and particularly characterized by a higher relative abundance of nitrogen-containing ions, especially in the methanol extract. Oxygen being involved in acidic functional groups such as carboxyls, hydroxyls, the profiles are directly impacted by solvent polarity in ESI(-) and show a higher number of oxygen-rich molecules when moving to polar extraction solvents (Figs. S4 and S5). In addition, organosulfurs are more oxygenated toward polar solvent reflecting the extractability of molecules with sulfonations (-$SO_3$) compared to less oxygen bearing compounds bearing sulfidic functional groups (-SH). Only the methanol extract showed composition with multiple heteroatoms.

Data mining of the molecular atlas showed that the most abundant mass differences were found to be equivalent to methylation ($CH_2$), hydration ($H_2O$), hydrogenation ($H_2$), oxygenation (O), sulfurizations ($SO_4$, $SO_3$, $SO_2$, SO, S), nitrogenations (HCN, NH, $NH_3$) as presented earlier with Murchison CM2 as well[1]. The search for removal or addition of these chemical functionalities' exact masses between all pairs of 23,100 molecular formulas resulted in the reconstruction of a mass difference network (MDiN) covering 154,578 pairs (edges) from among 19,675 molecular formulas (nodes). The remaining molecular formulas

could not be connected using these mass transitions. The most abundant mass difference transformations ($CH_2$, $H_2$, O and $H_2O$) amounted to 47% of all detected mass differences, connecting 20,232, 19,349, 17,111, and 16,620 pairs of compositions, respectively. Nitrogen (HCN, NH, $NH_3$) and sulfur-based mass differences (S, SO, $SO_2$, $SO_3$, $SO_4$) accounted for 22.6% and 20.0% of all transformations found, respectively. These structural continuum leads to differential solvent solubility and ionization efficiencies in the various ionization modes in FTICR-MS (Fig. S7).

[1]H NMR spectra of A0106 methanol extract provided quantitative assessment of key proton and carbon chemical environments, and showed distinct groups of narrow resonances from $\delta_H$ ~ 0.5-8.5 ppm in which alkyl branching and termination by methyl and carboxylic groups were the defining molecular features, followed by lower abundance of aliphatic OC**H** units, olefinic (coarsely dH <6.5 ppm) and aromatic unsaturation (coarsely dH <7 ppm) (Fig. 2). Olefins were incorporated in alkyl systems whereas aromatic $C_{ar}$**H** units likely comprised polycyclic aromatic rings with ill-constrained admixture of aromatic carboxylic acids and nitrogen heterocycles. Ammonium ($^{14}NH_4^+$) showed a distinct 1:1:1 triplet at $\delta_H$ ~ 4.5 ppm with $^1J_{NH}$ = 43 Hz and 1.2% relative abundance, exceeding other meteorite extracts. Ratios of methyl ($\delta_H$ < 1 ppm), non-methyl alkyl ($\delta_H$ ~ 1.0-1.65 ppm), and protons proximate to carboxylic acids HOOC-CHα-CHβ- (C**H**α: $\delta_H$ ~ 2.15 ± 0.15 ppm; C**H**β: $\delta_H$ ~ 1.65 ± 0.5 ppm) defined connectivity networks in two dimensional NMR spectra (Fig. S9) and provided reconstruction of average features of aliphatic branching in the

methanol extract with minor contributions from aliphatic oxygenation and olefinic unsaturation in A0106 SOM.

The dominance of hydrous silicate minerals (i.e., saponite, serpentine) and the absence of chondrules, already oriented the classification of the return material close to CI-type meteorites like Orgueil[6]. We compared the methanol extract of the sample A0106 with the methanol extracts of Murchison CM2 and Orgueil CI, two well-studied meteorites representative of their chondritic classes (Fig. 3). All extracts show full mass patterns over a range up to m/z 600. The van

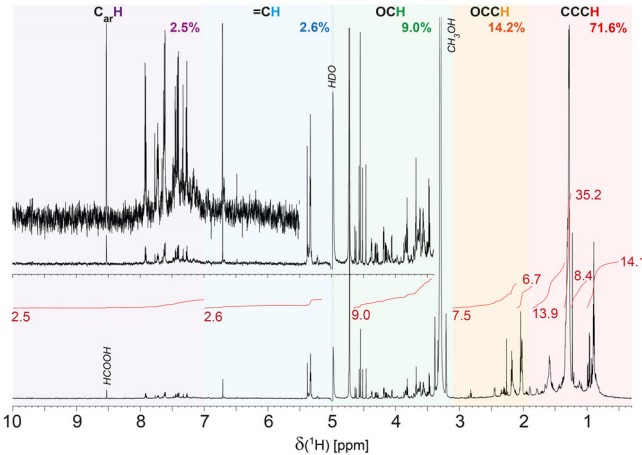

**Fig. 2 | Nuclear magnetic resonance spectroscopy of A0106.** [1]H NMR spectrum (800 MHz, CD$_3$OD) of mHDOM extract 4A0106, with [1]H NMR section integrals of main substructures (cf. Table S3). Apart from certain sharp NMR resonances which possibly denote individual aliphatic CHO molecules, mHDOM comprised relevant background of broad bulk [1]H NMR resonances that represent a huge diversity of aliphatic branched alkyl and remotely oxygenated aliphatic groups ($\delta_H$ ~ 0.6-2.7 ppm) and aliphatic OC**H** molecules ($\delta_H$ ~ 3.2-4.2 ppm); the overall curvature suggests higher relative abundance of aliphatic alcohols and ethers ($\delta_H$ < 3.8 ppm) in comparison with e.g. aliphatic esters ($\delta_H$ > 3.8 ppm). Ammonium ([14]NH$_4^+$) showed a distinct 1:1:1 triplet at $\delta_H$ ~ 4.5 ppm with [1]$J_{NH}$ = 42.7 Hz (1.2% of [1]H NMR integral; Fig. S8a–c).

Krevelen diagrams of A0106 show the same regular pattern as observed with Murchison that correspond to homolog chemical series but with lower abundance towards higher m/z. Orgueil showed the lowest abundance in mass signals and covers the same range in oxygen-rich region as A0106. Mass difference analysis normalized on CH2-content (Fig. 1E) showed a closest similarity in functionalization of A0106 with Orgueil CI than with Murchison CM2. Both A0106 and Orgueil share the same profile in the abundance of oxygenated hydrocarbon and organosulfur compounds. Murchison and A0106 share the same profile in heteroatom abundance, Orgueil having more molecules with many sulfur atoms.

Comparing the [1]H NMR spectra of the methanol extracts of A0106 with Murchison CM2 meteorite[19] revealed major distinction. [1]H NMR resonances were much narrower in A0106 than those in Murchison, indicating faster transverse (T$_2$) relaxation in Murchison extracts. This reflects the relative proportions of background bulk NMR resonances across the entire shift range $\delta_H$ in Murchison exceeding that of A0106. This indicates a larger chemodiversity of low abundance atomic environments in Murchison. Considerable background [1]H NMR resonances of pure aliphatic units (CCC**H**; $\delta_H$ ~ 0.7-2.5 ppm; Fig. S8B) probably arose from significant contributions of alicyclic rings in Murchison extracts.

To explore the high-dimensional dataset, we computed a principal component analysis (PCA) using SIMCA 13.0.3.0 (*Umetrics*) based on the intensities of all annotated m/z of the CHNOS compositional space unit-variance (UV) scaled and log transformed (Fig. 4A). We divided the data in a work (25 meteorite samples) and prediction set (11 meteorite samples) as shown in Table S1. The 36 CCs were chosen as to reflect a range of warm to low-temperatures history based on their thermal index from previous studies or known to have experienced strong water alteration and hydro-thermalism[22]. The work and prediction set were chosen randomly out of the 36 to check for the validity of the model. The variance absorbed by the first two valid components[23] is $R^2X$(cum)=0.51. Additionally, the prevision ability of the model was measured by the $Q^2$ index ($Q^2$(cum)=0.39). The PCA score scatter plot (Fig. 4A) differentiates the chemical space in two main regions. In one side, the *type-TI* CCs having seen low-temperature

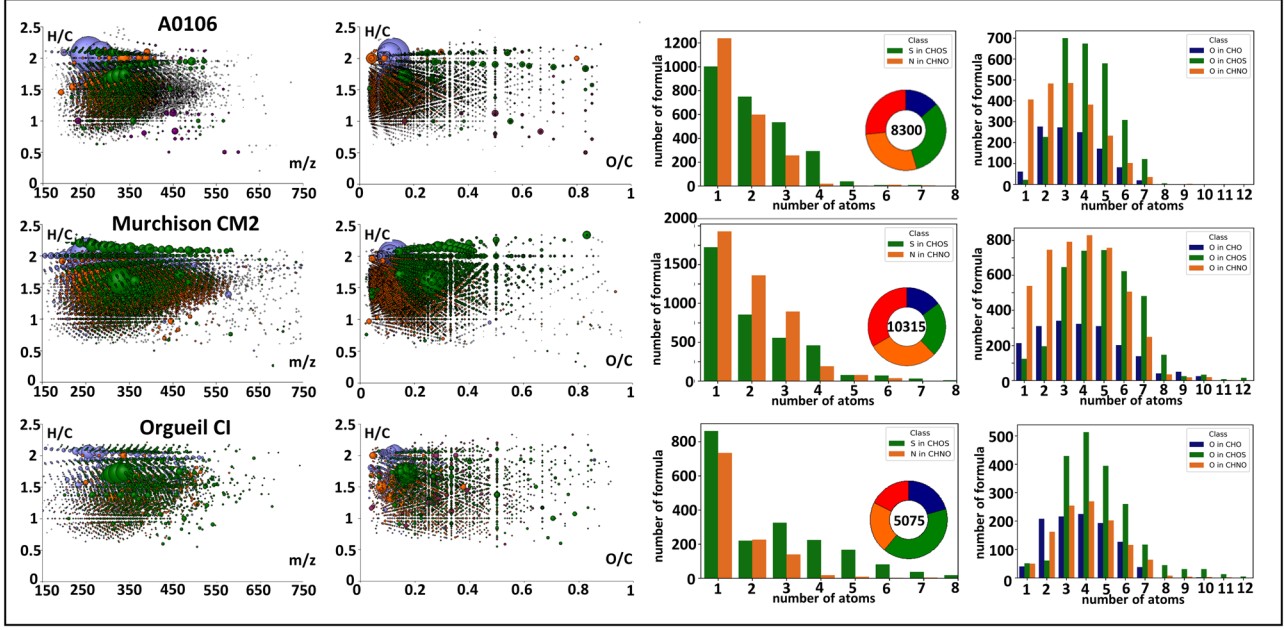

**Fig. 3 | FT-ICR-MS data evaluation of sample A0106 compared to carbonaceous meteorites.** Comparison of the mass edited H/C ratios and van Krevelen diagrams of sample A0106 with Murchison CM2 and Orgueil CI. The bubble size is proportional to the intensity in the original ESI(-)-FTICR-MS mass spectra and the color codes correspond to defined chemical classes of CHO(blue), CHNO(orange), CHOS(green) and CHONS(red). Histograms on the right are the abundance profiles of the heteroatoms as well as the oxygenation profiles of the CHO, CHNO and CHOS chemical families.

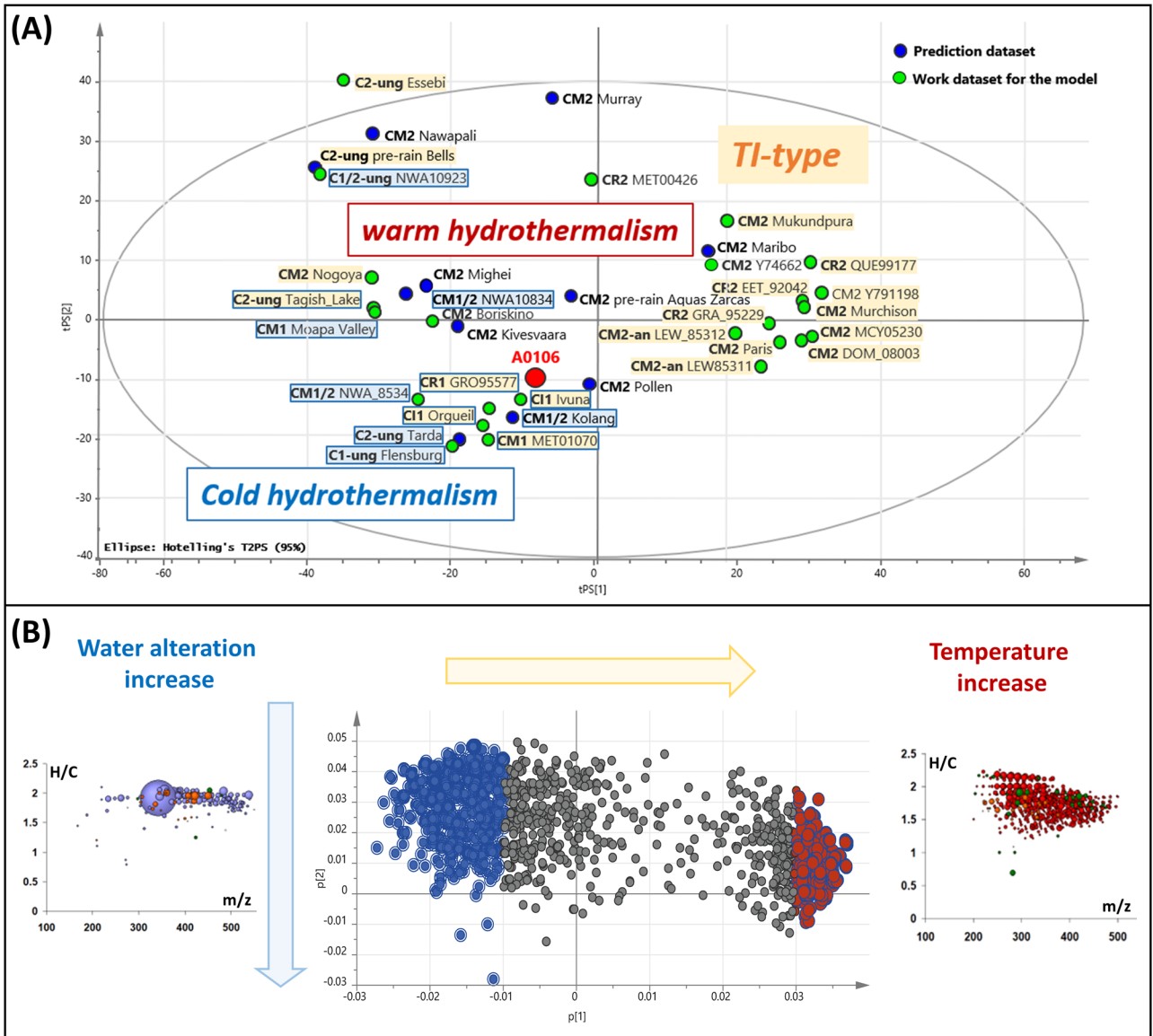

**Fig. 4 | Hydrothermal prediction of sample A0106 based on meteoritic soluble organic matter data. A** Principal component analysis (PCA) and model positioning the sample A0106 within 36 carbonaceous chondrites of TI-type (we defined the TI-type in Quirico et al. as a combination of XRD/mineral observation and Raman/IR data). All TI meteorites experiences aqueous alteration at temperatures <300 °C. In green the work dataset of the samples taken for the model and in blue the simulated positions form predicted dataset. **B** The loading plot represents the m/z values that differentiates an increase in water alteration (left) and temperature (right) represented in mass edited H/C ratios, accordingly; the bubble size is proportional to the intensity in the original ESI(-)-FTICR-MS mass spectra and the color codes correspond to defined chemical classes of CHO(blue), CHNO(orange), CHOS(green) and CHONS(red).

in various water alterations and in the other side, the chondrites of CM1, CR1, C1/2-ung, CM1/2, CI types that saw much water in their parent body history (*hydro-thermalism*). The most relevant masses of these two groups could be extracted based on their highest loading values and a representation in van Krevelen diagrams Fig. 4B.

The sulfurization of nitrogen compounds is mostly abundant at slightly increased temperatures reached in the typical TI-type meteorites[24]. Lower peak temperatures and higher water alteration corresponding to cold hydro-thermalism show higher abundance of saturated/mono-unsaturated long chained aliphatic carboxylic acid (Fig. 4B).

### Implication of chemical diversity on processes on the parent body

All solvent extracts of the A0106 surface sample from Ryugu showed very high molecular diversity in the mass range from 120 to 600 atomic mass units. Compounds containing nitrogen and sulfur were preferentially extracted in a solvent polarity gradient based on their chemical functionality (leading to continuum in polarity) and were detected as a function of their ionization specificity in ESI(−), ESI(+) or APPI(+) accordingly (Figs. S4 and S5). The molecular formula were observed in a continuum of oxygenation degree and relative abundances and corresponded to chemical families such as CH, CHO, CHN, CHS, CHOS, CHNO, CHNOS, CHONa, CHNONa, CHOSNa and CHNOSNa. The visualization of this chemical structural continuum of complex kerogen-like organic matter (OM) of the Ryugu surface sample was especially possible with Mass Difference Networks (MDiNs) that were reconstructed from mass spectral *m/z* values or theoretical masses after computation of molecular formulas (Figs. 2 and S7). The extreme molecular richness of these extracts reflects the high importance of chemical processes leading to this diversity involving the specific chemistry of nitrogen and sulfur[25]. We recently

linked the nitrogen compounds content between the carbonaceous meteorite Murchison and ices of dense molecular clouds and showed that the post-aqueous products of these organic residues produced from interstellar ice analogs share compositional similarities with SOM of the Murchison meteorite[26]. In particular, the nitrogen containing molecules in some CCs may originate from organic-rich ices, inherited from the dense molecular cloud of our solar system. Hydrothermal processing of this material may release ammonium ions as seen in the A0106 sample. A suite of amino acids and aliphatic amines were recently analyzed in the same A0106 samples[27] and showed different profiles than in CI, indicating multiple formation mechanisms on Ryugu's parent body. More recently, nucleobases were described in various Murchison, Murray and Tagish Lake meteorites showing that some of these nitrogen-rich photochemical derivatives produced in the interstellar medium could have been incorporated into asteroids during solar system formation[28].

We did not observe any organic compounds containing magnesium (i.e. CHOMg, CHOSMg), which were described in various CCs with increasing temperature history. The absence of organomagnesium compounds was recently shown in the recent C1-ung fall Flensburg[29] and in the C2-ung Tarda[30], two meteorites that also saw high degree of water alteration. These characteristics reflect the low temperature hydrothermal processing (<-150 °C) involved on the parent body[2,3], as well as the severe aqueous alteration and hydrolysis of the whole system. These results on the SOM are in line with the reported studies on insoluble organic matter (IOM) showing that Ryugu's organic material was modified by aqueous alteration on the asteroid parent body[31]. Considering the abundance profile of thousands of oxygenated molecular formula of the methanol extract of 36 meteorites we were able to dress a predictive A0106 being highly water processed at low temperature such as observed in C2-ung Tarda, C1-ung, CM1/2, CM1, CR1 or CI (Fig. 4).

Ryugu's molecular atlas is characterized by a continuum in oxidized heterocyclic organosulfur and organonitrogen compounds corresponding to a molecular array snapshot of ongoing abiotic reaction processes on the parent body, heavily impacted by water alteration. Abiotic molecular synthesis was described recently in the Martian meteorite Tissint and earlier in ALH 84001 as being the origin of the complex OM observed co-localized to evolving minerals[32,33]. Here as well, the chemical signature of the SOM may result from a complex history in carbonization and serpentinization, water-rock interactions generating and consuming water during olivine and pyroxenes dissolution and leaving behind serpentine and an array of oxygenated novel carbon species[34]. Co-localization of carbon with mineral phases and coevolution within a redox homeostasis will oxidize newly synthesized hydrocarbons while mineral and/or organic electron donors in the mixtures will reduce[32]. These arrays of mixed and localized reactions will create further complex organic molecules in a structural continuum in mass, oxygenation, carbon saturation and heteroatom content. Our results from both ESI- and APPI-FTICR-MS show the abundance of PAHs and hydrocarbons (CH) in a wide molecular range as well as their oxygenation products (CHO). While a hydrogenation process may favor dearomatization i.e. of PAHs, water alteration and hydrolysis processes would rather lead to more polar oxygen-rich compounds to reach a thermodynamically stable steady state. Serpentinization processes are thermodynamically favored, exothermic and therefore can also generate local low heat of up to only 150–200 °C on small parent bodies, thereby activating an additional chemical process of carbonization to again generate oxidized carbon and water[35]. Finding no organomagnesium compounds also confirms the high water alteration and hydrolysis processes affecting the organic matter in the system as essential processes in organic evolution on Ryugu. The two described geochemical processes may be thermodynamically coupled and involve closely the mineral and organic phases within their complex coevolution and both occur during the water-rock interaction. In this way, sulfurization processes may also occur involving the reactive sulfur species generated during sulfur mineral evolution. These processes are described in terrestrial environments and the incorporation of reduced inorganic sulfur (S) into OM in anoxic environments is considered a vital pathway by which OM is preserved for millions of years and the iron sulfide precipitation is believed to compete with OM preservation by sulfurization[36]. The sulfur chemistry continuum in CHOS and CHNOS observed in sample A0106 reflects similar abiotic processes on Ryugu parent body.

The high molecular diversity of SOM in the Ryugu sample A0106 was comparable to meteorite SOM having seen high water alteration processes at low temperatures. The methanol-extracted SOM profile are consistent with A0106 being closely related to CM1/2, CI or CM1 type of meteorites. The presence of homologous chemical series in nitrogen-containing organics may support an origin from organic ices, inherited from the dense molecular cloud of our solar system. The molecular atlas of Ryugu consists of a continuum in molecular size, oxygenations, and sulfurization of both hydrocarbons and nitrogen-containing organics and reflect a complex series of multiple and successive sources in abiotic organic synthesis possibly involving redox-based water-rock reaction, water dissociation (e.g. water radiolysis), with subsequent serpentinization carbonation and further organic solvolysis. Low amounts of organomagnesium confirmed low processing temperatures around 150 °C maximum in accidence to serpentinization processes. Thus, the SOM of Ryugu C-type asteroid surface is a source of organic diversity and complexity and may be regarded as a cradle of evolving prebiotic molecules in the Solar system, a source of molecular precursors of life itself.

## Methods

### Meteorite samples used for analysis and structural comparison
The list of the meteorites analyzed in the study and used for the model in Fig. 4 is given in Table S1.[37,38]

### Sample description
The Ryugu sample extracted was sample A0106, the surface samples stored in Chamber A. It was extracted sequentially with solvents of increasing polarity, starting with hexane, dichloromethane, methanol and water to generate/isolate chemical fractions of SOM with molecules of increasing polarity. This was done at JAXA and samples were distributed to various Labs for analysis as described in ref. 5.

### Fourier transform ion cyclotron resonance mass spectrometry
The experimental study was performed on a high-field FT-ICR mass spectrometer from Bruker Daltonics with a 12-T magnet from Magnex[1]. A time domain transient with 4 MWords was obtained and Fourier-transformed into a frequency domain spectrum. The frequency domain was afterward converted to a mass spectrum by the SolariX Control program of Bruker Daltonics. The ion excitations were generated in broadband mode (frequency sweep radial ion excitation) and 3000 scans were accumulated for each mass spectrum in a mass range of 147–1000 amu. Ions were accumulated for 300 ms before ICR ion detection. The pressure in the quadrupole/hexapole and ICR vacuum chamber was $3 \times 10^{-6}$ and $6 \times 10^{-10}$ mbar, respectively. For CID-MS/MS, ions were accumulated for 3 s.

The ESI source (Apollo II; Bruker Daltonics) was used in negative ionization mode and the APPI source in positive mode. The methanolic solutions were injected directly into the ionization source by means of a microliter pump at a flow rate of 120 $\mu$L h$^{-1}$ in ESI and at 500 $\mu$L h$^{-1}$ APPI. A source heating temperature of 200 °C was maintained and no nozzle-skimmer fragmentation was performed in the ionization source. The instrument was previously externally calibrated by using arginine negative cluster ions (5 mg L$^{-1}$ arginine in methanol).

FT-ICR mass spectra with *m/z* from 95 to 1000 amu were calibrated externally and internally to preclude alignment errors. Subsequently, the mass spectra were exported to peak lists at a signal-to-noise ratio ≥3. Elemental formulas were calculated combinatorically within a mass accuracy window of ±0.2 ppm for each peak in batch mode by an in-house software tool and validated via the senior-rule approach/cyclomatic number, assuming valence 2 for S and valence 4 (coordination number) for Mg[2].

FT-ICR-MS analysis enables highly resolved ($R > 10^6$ at *m/z* 200) and accurate chemical mass analysis of electrospray generated ions within a 200 ppb error window over a wide mass range from *m/z* 100–1000. The weight of the ions is measured with a precision lower than the mass of an electron ($\Delta m/z = 0.0003\ amu$) and the specific signals can be differentiated with the same mass precision due to the ultrahigh resolution. These exact masses of the ions can routinely be converted into unique compositional formula bearing the light elements C-, H-, N-, O-, S-, Mg (or any other element in target), also taking account of their possible Cl and Na adducts and of their natural isotopic abundance.

The solvent extracts generated thousands of individual signals that were converted into elementary compositions (formula); these are all represented in van Krevelen type of diagrams (H/C vs. O/C) or related (H/C vs *m/z*) in which each formula is represented by a dot (the size of the dot is proportional to its abundance) as a projection of the relative oxygenation degree (O/C) and saturation degree (H/C) for various classes of compound types (CHO, CHNO, CHOS, CHNOS, CH, CHS, CHN, and selected Na$^+$-adducts). Aromaticity equivalents Xc were calculated as reported earlier[39] and plotted as in Fig. S6.

### Nuclear magnetic resonance spectroscopy

A Bruker Avance III spectrometer and TopSpin 3.5/PL7 software were used to acquire nuclear magnetic resonance (NMR) spectra of aqueous extracts of Hayabusa methanolic extracts of which ~30 µg were exchanged with CD$_3$OD (99.96 % $^2$H, Aldrich) three times on a vacuum line[19]; after sealing in a 1.7 mm Bruker match tube, the sample was centrifuged without visible solid. Murchison extract A5[19] was used here for comparison to sample A0106.

A cryogenic inverse geometry 5 mm *z*-gradient $^1$H/$^{13}$C/$^{15}$N/$^{31}$P QCI probe ($B_0 = 18.8$ T) was used for 1D $^1$H NMR and proton-detected 2D NMR spectra. Transmitter pulses were at ~10 µs for $^1$H and $^{13}$C. The one bond coupling constant $^1$J(CH) used in 2D $^1$H,$^{13}$C DEPT-HSQC spectra (*hsqcedetgpsisp2.2*) was set to 145 Hz; other conditions: $^{13}$C 90 deg decoupling pulse, GARP (70 µs); 50 kHz WURST 180 degree $^{13}$C inversion pulse (Wideband, Uniform, Rate, and Smooth Truncation; 1.2 ms); F2 ($^1$H): spectral width of 11160.7 Hz (13.95 ppm); 1.25 s relaxation delay; F1 ($^{13}$C): SW = 36052 Hz (180 ppm). HSQC-derived NMR spectra were computed to a 8192 × 1024 matrix. Gradient (1 ms length, 450 µs recovery) and sensitivity enhanced sequences were used for all 2D NMR spectra. Absolute value COSY, and phase sensitive echo-antiecho TOCSY spectra (*cosygpmfppqf, dipsi2etgpsi*) used a spectral width of 9615.4 Hz and were computed to a 16,384 × 2048 matrix; other NMR acquisition conditions are given in Table S2.

### Data availability

All data from the mission are available at the DARTS archive www.darts.isas.jaxa.jp/planet/project/hayabusa2/ and on the Small Bodies Node of the NASA Planetary Data System https://pds-smallbodies.astro.umd.edu/data_sb/missions/hayabusa2/. The samples of Ryugu are curated by the JAXA Astromaterials Science Research Group; distribution for analysis is through an Announcement of Opportunity available at https://jaxa-ryugu-sample-ao.net. The FTICR-MS raw data can be make available in contacting the corresponding author and can be adapted depending on further utilization.

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

## Acknowledgements

The Hayabusa2 project has been led by JAXA (Japan Aerospace Exploration Agency) in collaboration with DLR (German Space Center) and CNES (French Space Center), and supported by NASA and ASA (Australian Space Agency). We thank all of the members of the Hayabusa2 project for their technical and scientific contributions. We thank Dr. Laurence Garvie from ASU (Arizona State University) and the Buseck Center Meteorite Studies for some selected fragments of meteorites used in this study. This research is partly supported by the Japan Society for the Promotion of Science (JSPS) under KAKENHI grant numbers JP20H00202, JP20H05846, JP20K20485, JP20K14549, JP21J00504, JP21H01203, JP21H04501, and JP21KK0062. J.P.D., J.C.A., E.T.P., D.P.G., H.L.M., J.E.E., and H.V.G. are grateful to NASA for support of the Consortium for Hayabusa2 Analysis of Organic Solubles. This research is also funded by the Deutsche Forschungsgemeinschaft (DFG, German Research Foundation) – Project-ID 364653263 – TRR 235 (CRC 235).

## Author contributions

P.S.-K. designed this research. P.S.-K. and N.H. conducted experiments and analyzed data in cooperation with M.H., M.L., F.M., L.B., and E.Q. P.S.-K. wrote the paper. S.T., H. Yurimoto, T. Nakamura, T. Noguchi, R.O., H. Yabuta, and H.N. administered the initial analysis with J.D., Y. Takano, and M.A. T. Yada, M.N., K.Y., A.N., M.Y., A.M., and T.U. curated samples. K.S., T.O., S.N., F.T., S. Tanaka, and T.S. contributed to science operations of the spacecraft. S.-i.W. and Y. Tsuda administered the project. All authors discussed the results and commented on the manuscript.

## Funding

## Competing interests

The authors declare no competing interests.

## Additional information

## Hayabusa2-initial-analysis SOM team

Philippe Schmitt-Kopplin [1,2,3 ✉], Norbert Hertkorn[2], Hiroshi Naraoka [7], Yoshinori Takano [5], Jason P. Dworkin [6], Kenji Hamase[15], Aogu Furusho[15], Minako Hashiguchi[14], Kazuhiko Fukushima[16], Dan Aoki[16], José C. Aponte[6], Eric T. Parker[6], Daniel P. Glavin[6], Hannah L. McLain[6], Jamie E. Elsila[6], Heather V. Graham[6], John M. Eiler[17], Alexander Ruf[18], Francois-Regis Orthous-Daunay[4], Junko Isa[19], Véronique Vuitton[4], Roland Thissen[20], Nanako O. Ogawa[21], Saburo Sakai[21], Toshihiro Yoshimura[21], Toshiki Koga[21], Haruna Sugahara[9], Naohiko Ohkouchi[21], Hajime Mita[22], Yoshihiro Furukawa[10], Yasuhiro Oba[23] & Shogo Tachibana [8,9]

[15]Graduate School of Pharmaceutical Sciences, Kyushu University, Motooka 744, Nishi-ku, Fukuoka 819-0395, Japan. [16]Graduate School of Bioagricultural Sciences, Nagoya University, Chigusa-ku, Nagoya 464-8601, Japan. [17]Division of Geological and Planetary Sciences, California Institute of Technology, Pasadena, California 91125, USA. [18]Excellence Cluster ORIGINS, Garching 85748, Germany. [19]Earth-Life Science Institute (ELSI), Tokyo Institute of Technology, Meguro-ku, Tokyo 152-8550, Japan. [20]Université Paris-Saclay, CNRS, Institut de Chimie Physique, UMR8000, 91405 Orsay, France. [21]Biogeochemistry Research Center (BGC), Japan Agency for Marine-Earth Science and Technology (JAMSTEC), 2–15 Natsushima, Yokosuka 237-0061, Japan. [22]Department of Life, Environment and Applied Chemistry, Fukuoka Institute of Technology, Higashi-ku, Fukuoka 811-0295, Japan. [23]Institute of Low Temperature Sciences (ILTS), Hokkaido University, Kita-ku, Sapporo 060-0810, Japan.

