## [Peer Review File · Nature Communications]

Soluble organic matter Molecular atlas of Ryugu reveals cold hydrothermalism on C-type asteroid parent bodyReviewer #1 (Remarks to the Author):

The Hayabusa2 mission successfully returned samples from the surface and sub-surface of the carbonaceous near-Earth asteroid Ryugu. Analysis of these samples allows us to investigate the initial conditions in the Solar System and the formation and evolution of planetary bodies. Crucially, the samples are from a known body and have experienced minimal terrestrial contamination, providing an opportunity to characterise pristine organic matter from the earliest history of the Solar System.

This is a generally well-written manuscript that reports on some impressive measurements and provides a detailed description of organic species in a Ryugu sample. The results will certainly be of interest to researchers in many fields – carbonaceous asteroids like Ryugu likely delivered pre-biotic molecules and other volatile species to Earth – but the manuscript has several short-comings that should be addressed before it is accepted for publication. In particular: -

- the Introduction doesn't clearly set out why this research is important or what hypothesis is being tested by these measurements.

- there is no description of the sample. Perhaps this is available in other papers, but as Ryugu samples vary in mineralogy and degree of aqueous alteration (e.g., Nakamura et al. (2022) Science), some information about the analysed particle should be included.

- The Discussion doesn't really place these results in the context of what is already known about the formation and evolution of Ryugu. For example, how do these data fit with the recent organic results and interpretation from Yabuta et al. (2023) Science, Naraoka et al. (2023), Oba et al. (2023) Nat. Comms., and Parker et al. (2023) GCA, and the aqueous and thermal history of Ryugu inferred from mineralogical studies?

Summary

P2 L1-2: second half of this sentence isn't very clear and should be re-written.

P2 L5: maybe context rather than light here?

P2 L7: C-type asteroid hasn't been defined

P2 L11: "Results showed..."

P2 L13: think it needs to be clear that CI chondrites are meteorites. It might not be obvious to a non-specialist reader.

Main Text

P3 L6: at the time - samples were also returned from the Moon by Chang'e 5 in December 2020

P3 L16: either CI or CI1 chondrites. Recommend that you don't use both because it confuses non-specialists.

P3 L17-19: this sentence is really unclear. As written, it sounds like C, H, N, O and S were the only elements measured in Ryugu samples, which isn't the case. Also, those elements are mainly present in carbonate and sulphide minerals, and organic matter.

P3 L19-21: this statement needs a reference!

P3 L22: don't need to repeat that Orgueil is a CI chondrite. But you might want to mention somewhere that the CI chondrites are all falls.

P3 L24: suggest, "...aminovaleric acid in the Ryugu samples indicated a higher peak temperature than the CI chondrites as these compounds are typical in CV and CO

chondrites processed at elevated temperatures of up to ~300°C.”

P3 L26-28: needs a reference. Also, not clear why these particular species are important?

P3 L29: define FTIR.

Also, do you mean FTIR spectra of extracted organic matter (i.e., not individual particles)? And the statement needs a reference.

P3 L36-37: is that surprising? Murchison is a CM chondrite (which isn't mentioned in the text) from a different parent body. More relevant is how the profile of these species compares to the CI chondrites?

Molecular Atlas

P4 L1: what was the sample size/mass? Was any characterisation done before the SOM was extracted? Ryugu samples contain several lithologies with different degrees of aqueous alteration. Also, was the sample exposed to air prior to extraction? The Ryugu grains are very sensitive to terrestrial contamination and modification on short timescales.

P4 L7: define NMR

P4 L17: what is meant by “fresh” CCs? Aguas Zarcas is a recent fall, but Murchison fell >50 years ago and I wouldn't consider it “fresh” because these rocks are so reactive. And what was the curation history of the analysed samples? For example, Aguas Zarcas samples were collected both before and after rainfall in the area.

P4 L22: guessing this should be Table S1? It's not clear from the text if all the samples listed in Table S1 were analysed as part of this study.

P4 L23: if this is a result of this study, why is an LPSC abstract being cited here?

P4 L24: mass

P4 L37: Figure 1 shows...

P6 L8: which Fig. 6 is this referring to?

P7 L8: two well-studied meteorites

P7 L11: ...show the same...

P7 L29: ...reflects the relative...

Fig. 3 caption: ...are the abundance profiles...

P8 L3: 25 + 11 = 36 CCs? There are 36 listed in Table S1.

P8 L9: suggest deleting “nicely” and differentiates

P8 L10: this is quite confusing to read. To me it looks like Fig. 4a broadly divides the meteorites into less altered type 2's and the more altered type 1's, although there is a reasonable number of type 2 meteorites on the left side of the plot. Interestingly, the CM2's on the left all appear to be falls (and predicted datasets) – does that suggest a difference between falls and finds?

P8 L21: what evidence is there for increased temperatures in the type 1 meteorites?

P9 L24: delete exceptionally

P9 L30: Flensburg is a C1ung, not a CI chondrite

P9 L31: "heavy water" is confusing as it could refer to the isotopic composition of the water. I think you mean that these meteorites experienced a high degree of aqueous alteration.

Tarda is a type 2 breccia and contains lithologies that were not fully altered. I'm not aware of any estimates for peak temperatures for Tarda, but if it's similar to Tagish Lake then it's likely to contain lithologies that were altered at temperatures >150°C.

P9 28-36: This paragraph describes the properties of Flensburg (C1ung) and Tarda (C2ung), but then concludes in a single line that Ryugu is most like a CI chondrite. There should be a more detailed comparison of Ryugu to the CI chondrites. Also, based on Fig. 4a, I'd argue that Ryugu is just as similar to highly CR1 or CM1 chondrites as the CIs.

The statement that CI chondrites were highly altered at low temperatures needs a reference, and also seems to contradict the earlier claim that Type 1 meteorites were altered at slightly elevated temperatures (P8 L21).

P9 L38: don't think atlas needs to be capitalised

P10 L3-6: this statement needs references to support it

P10 L7: ...hydrolysis processes affecting...?

P10 L11-18: the Ryugu samples contain beautiful, euhedral, and often large, grains of pyrrhotite that formed during aqueous alteration. The mineralogy of Ryugu is now well-constrained and this should be considered in the discussion of organic synthesis here.

P10 L21: ...processes at lower temperatures. Lower than what?

P10 L22: ...are consistent with A0106 being closely related to...

I agree that the Ryugu samples are similar to CI1, CM1 (and also CR1) meteorites, but this appears to slightly differ from the claim that they are like CI chondrites (P9 L36).

P10 L30: mineralogical evidence suggests that Ryugu was altered at temperatures <100°C, possibly even <50°C

Supplementary Materials

I have several issues with Table S1:-

Flensburg is a C1ung

Where are the weathering grades from? Just because something is a fall, doesn't make it a weathering grade 0. For example, Ivuna and Orgueil contain abundant, terrestrially formed sulphates.

The two columns describing the degree of alteration / petrologic type need to be explained and are not up-to-date, incomplete, or incorrect. The best starting point for the petrologic classifications is Rubin et al. (2007) GCA 71:2361 (not the Rubin (2007) that is currently cited), while it should also be noted that many of these meteorites are breccias with lithologies altered to different degrees (e.g. Paris is a 2.9 – 2.6). Rubin didn't look at the CI chondrites and calling them 1 doesn't make sense in this column. Also, petrographic classifications have been reported for other meteorites e.g., Aguas Zarcas, Maribo.

The Alexander et al. (2013) classification scheme is based on bulk H and C abundances (not petrography). Again, some data is missing (e.g., Flensburg) and it should be made clear in the caption that a 2.0 in the Rubin scale is not the same degree of alteration as a 2.0 in the Alexander scale. Not to confuse things further, but a note that bulk phyllosilicate abundances have been reported for meteorites such as Moapa Valley and Mukundpura, where no petrologic classification is given in Table S1.

The caption needs to briefly explain what TI means. This is taken from Quirico et al. (2018), but is really a combination of XRD/mineral observations of Nakamura (2005) and the Raman/IR data of Quirico et al., to estimate peak temperatures for carbonaceous chondrites. The TI meteorites experienced low temperature (<300°C) aqueous alteration and do not record any post-hydration metamorphism (i.e., TII – IV in Quirico et al.). There is published XRD data for many of the samples in Table S1 that is consistent with them being TI (e.g., Howard et al. 2015 GCA 149:206, King et al. 2017 M&PS 52:1197).

Fig. S1 caption: Orgueil should be CI1 (if using CM2 for Murchison). Also, typo – green.

Fig. S2 caption: The bubble size expresses... ..and the colors correspond... ..One clearly observes...

Also, define n.t. in the figure captions.

Fig. S3 caption: m/z expanded? ...One clearly observes...

Fig. S8a: either ...an enhanced molecular interaction... or ...enhanced molecular interactions...

...A0106 showed a larger proportion of...

Reviewer #2 (Remarks to the Author):

This study highlights the high diversity of Ryugu organic matter using mass spectrometry and NMR spectroscopy. The results are interesting (though redundant with existing literature) and worth publishing in Nature Communications. I made comments, hereafter provided line by line, that I think will help the reader better understand.

Page 2, line 3 : Please rephrase “since the parent body”.

4-7 : Please remove “systematically” and explain why you couldn’t perform ESI(+) on hexane extract and APPI(+) on water extract.

4-32 : What S/N did you use to reduce the 200,000 signals ? What type of information have you possibly lost ?

4-22 : Table x is Table S1 ?

Figure 1 (E) : Please invert SO and S columns.

8-3 : “moderate” in the text corresponds to “warm hydrothermalism” in the figure 4 (A) ?

Figure 4 (A) : How exactly did you chose the 2 parameters of the PCA ? The “warm hydrothermalism” annotation corresponds to which CCs ? Those in the upper left corner of the PCA ?

9-31 : Do you consider Tarda highly altered only because there is no organomagnesium compounds in it ?

10-16 : Please rephrase the last sentence of the paragraph.

11-5 : Please update Naraoka paper.

Table S2 : Missing figure names ?

Figure S1 : Please highlights both C17H35O3S and C20H31OS.

Typos :

Page 2, line 7 : shines

3-30 : PAH

4-24 : mass instead of masse

9-25 : various

10-7 : affection

10-15 sulfide

10-21 : Please, leave only one "seen".

10-26 : reflects

19-9 : green

RESPONSE TO THE REVIEWER'S COMMENTS.

Reviewer 1

Review of Molecular Atlas of Soluble Organic Matter of Ryugu Reveals Cold Hydrothermalism on C-type Asteroid Parent Body (NCOMMS-23-06104-T)

The Hayabusa2 mission successfully returned samples from the surface and sub-surface of the carbonaceous near-Earth asteroid Ryugu. Analysis of these samples allows us to investigate the initial conditions in the Solar System and the formation and evolution of planetary bodies. Crucially, the samples are from a known body and have experienced minimal terrestrial contamination, providing an opportunity to characterise pristine organic matter from the earliest history of the Solar System.

This is a generally well-written manuscript that reports on some impressive measurements and provides a detailed description of organic species in a Ryugu sample. The results will certainly be of interest to researchers in many fields – carbonaceous asteroids like Ryugu likely delivered pre-biotic molecules and other volatile species to Earth – but the manuscript has several short-comings that should be addressed before it is accepted for publication.

→ We thank the reviewer for his general impression.

In particular: -

- the Introduction doesn't clearly set out why this research is important or what hypothesis is being tested by these measurements.

→ We added the needed information at the end of the introduction

"The returned surface sample of Ryugu A0106 enabled now a direct analysis of SOM from the surface of the parent body and the comparison with meteoritic SOM. Here, we report the molecular characteristics of Ryugu's SOM from non-targeted organic analysis. More specifically we compared the ultrahigh-resolution mass spectroscopy analysis profile of the methanol extracts to 36 CCs that experienced moderate temperatures and partial aqueous alteration to position Ryugu's organic diversity in the context of possible hydrothermalism."

- there is no description of the sample. Perhaps this is available in other papers, but as Ryugu samples vary in mineralogy and degree of aqueous alteration (e.g., Nakamura et al. (2022) Science), some information about the analysed particle should be included.

→ The sample description was short in the main text. We added additional details and references needed in Supplementary materials.

- The Discussion doesn't really place these results in the context of what is already known about the formation and evolution of Ryugu. For example, how do these data fit with the recent organic results and interpretation from Yabuta et al. (2023) Science, Naraoka et al. (2023)[5], Oba et al. (2023) Nat. Comms.[27], and Parker et al. (2023) GCA, and the aqueous and thermal history of Ryugu inferred from mineralogical studies?

→ We added latest references in the discussion to confirm and crosslink the results in the context of Ryugu's aqueous and thermal history.

Summary

P2 L1-2: second half of this sentence isn't very clear and should be re-written.

→ We added the information related to the chemistry reflecting a history in the parent body

P2 L5: maybe context rather than light here?

→ Yes, we changed accordingly.

P2 L7: C-type asteroid hasn't been defined

→ It is defined in the introduction ... we rather took the space of the Abstract to focus on the results.

P2 L11: "Results showed..."

→ Yes, we changed accordingly.

P2 L13: think it needs to be clear that CI chondrites are meteorites. It might not be obvious to a non-specialist reader.

→ Yes, we changed accordingly.

Main Text

P3 L6: at the time - samples were also returned from the Moon by Chang'e 5 in December 2020

→ Yes, this was added.

P3 L16: either CI or CI1 chondrites. Recommend that you don't use both because it confuses non-specialists.

→ Typo was changed, thank you.

P3 L17-19: this sentence is really unclear. As written, it sounds like C, H, N, O and S were the only elements measured in Ryugu samples, which isn't the case. Also, those elements are mainly present in carbonate and sulphide minerals, and organic matter.

→ We changed the sentences accordingly to give the information on the elements C, H, N, O, S only due to their importance in this particular manuscript.

P3 L19-21: this statement needs a reference!

→ Done

P3 L22: don't need to repeat that Orgueil is a CI chondrite. But you might want to mention somewhere that the CI chondrites are all falls.

→ Added.

P3 L24: suggest, "...aminovaleric acid in the Ryugu samples indicated a higher peak temperature than the CI chondrites as these compounds are typical in CV and CO chondrites processed at elevated temperatures of up to ~300°C."

→ Exchanged, thank you

P3 L26-28: needs a reference. Also, not clear why these particular species are important?

→ Done

P3 L29: define FTIR.

Also, do you mean FTIR spectra of extracted organic matter (i.e., not individual particles)? And the statement needs a reference.

→ These results are from the ref. (5) added

P3 L36-37: is that surprising? Murchison is a CM chondrite (which isn't mentioned in the text) from a different parent body. More relevant is how the profile of these species compares to the CI chondrites?

→ The data was not compared to CI types in the literature, but showing it is different to CM2 also confirms the trend.

"CM2" was added to inform what Murchison is.

Molecular Atlas

P4 L1: what was the sample size/mass? Was any characterisation done before the SOM was extracted? Ryugu samples contain several lithologies with different degrees of aqueous alteration. Also, was the sample exposed to air prior to extraction? The Ryugu grains are very sensitive to terrestrial contamination and modification on short timescales.

→ This information was given in SI also with the main reference (5) describing in detail the procedures done by JAXA.

P4 L7: define NMR

→ Done

P4 L17: what is meant by "fresh" CCs? Aguas Zarcas is a recent fall, but Murchison fell >50 years ago and I wouldn't consider it "fresh" because these rocks are so reactive. And what was the curation

history of the analysed samples? For example, Aguas Zarcas samples were collected both before and after rainfall in the area.

→ Fully agree that the wording “fresh” is not the right one ... thanks for the notice ... we adapted the wording in the text.

P4 L22: guessing this should be Table S1? It's not clear from the text if all the samples listed in Table S1 were analysed as part of this study.

→ Done

P4 L23: if this is a result of this study, why is an LPSC abstract being cited here?

→ The LPSC abstract only showed preliminary results on the study as the proof of concept.

P4 L24: mass

→ done, thanks

P4 L37: Figure 1 shows...

→ done

P6 L8: which Fig. 6 is this referring to?

→ It's Fig. S7

P7 L8: two well-studied meteorites

→ done

P7 L11: ...show the same...

→ done

P7 L29: ...reflects the relative...

→ done

Fig. 3 caption: ...are the abundance profiles...

→ done

P8 L3: $25 + 11 = 36$ CCs? There are 36 listed in Table S1.

→ 😊 yes ! thanks, done

P8 L9: suggest deleting “nicely” and differentiates

→ done

P8 L10: this is quite confusing to read. To me it looks like Fig. 4a broadly divides the meteorites into less altered type 2's and the more altered type 1's, although there is a reasonable number of type 2

→ The data are interpreted based on the metadata available ... a general study is in validation with more than 100 meteorites confirming even more the differentiation we describe here.

meteorites on the left side of the plot. Interestingly, the CM2's on the left all appear to be falls (and predicted datasets) – does that suggest a difference between falls and finds?

→ Falls are distributed all over (Maribo, Paris, Mukundpura on the right i.e.) and we did not find any valid model to differentiate falls from finds.

P8 L21: what evidence is there for increased temperatures in the type 1 meteorites?

→ T1-Type meteorites were attributed according to our paper Quirico et al 2018 and are of slightly higher temperature than the C1, CM1 or CM1/2 studied herein.

P9 L24: delete exceptionally

→ Done

P9 L30: Flensburg is a C1ung, not a CI chondrite

→ changed

P9 L31: "heavy water" is confusing as it could refer to the isotopic composition of the water. I think you mean that these meteorites experienced a high degree of aqueous alteration. Tarda is a type 2 breccia and contains lithologies that were not fully altered. I'm not aware of any estimates for peak temperatures for Tarda, but if it's similar to Tagish Lake then it's likely to contain lithologies that were altered at temperatures >150°C.

→ I changed it to "high degree of"

In deed with breccia we always will have zones of differential alterations and problems of representative samples.

Tarda is in a blue box in the Figure 4 and such as Flensburg, Kolang and NWA 8534 was not used to build the model but fall in the "cold hydrothermalism" zone.

P9 28-36: This paragraph describes the properties of Flensburg (C1ung) and Tarda (C2ung), but then concludes in a single line that Ryugu is most like a CI chondrite. There should be a more detailed comparison of Ryugu to the CI chondrites. Also, based on Fig. 4a, I'd argue that Ryugu is just as similar to highly CR1 or CM1 chondrites as the CIs.

The statement that CI chondrites were highly altered at low temperatures needs a reference, and also seems to contradict the earlier claim that Type 1 meteorites were altered a slightly elevated temperatures (P8 L21).

→ I fully agree ... my results on SOM show high water processing and low temperatures ... The literature position the Ryugu samples close to CI. I thus rather describe the close profiles of Ryugu to the specific types in Figure 4 and adapted the text accordingly to:

"Considering the abundance profile of thousands of oxygenated molecular formula of the methanol extract of 36 meteorites we were able to dress a predictive A0106 being highly water processed at low temperature such as observed in C2-ung Tarda, CM1/2, CM1, CR1 or CI (Figure 4)."

P9 L38: don't think atlas needs to be capitalised

→ I adapted.

P10 L3-6: this statement needs references to support it

→ done

P10 L7: ...hydrolysis processes affecting...?

→ done

P10 L11-18: the Ryugu samples contain beautiful, euhedral, and often large, grains of pyrrhotite that formed during aqueous alteration. The mineralogy of Ryugu is now well-constrained and this should be considered in the discussion of organic synthesis here.

→

P10 L21: ...processes at lower temperatures. Lower than what?

→ Changed to "low"

P10 L22: ...are consistent with A0106 being closely related to...

→ done

I agree that the Ryugu samples are similar to CI1, CM1 (and also CR1) meteorites, but this appears to slightly differ from the claim that they are like CI chondrites (P9 L36).

→ Fully agree ! that's why I rather precised my close results in the SOM profiles to the whole group of meteorites in P9 L36 ... reflecting form the SOM-side the high degree of water alteration.

P10 L30: mineralogical evidence suggests that Ryugu was altered at temperatures <100°C, possibly even <50°C

→ Just to be on the good side I jept only 150°C

Supplementary Materials

I have several issues with Table S1:-

Flensburg is a C1ung

→ adapted

Where are the weathering grades from? Just because something is a fall, doesn't make it a weathering grade 0. For example, Ivuna and Orgueil contain abundant, terrestrially formed sulphates.

→ Right CI are rather hydrophilic and see secondary terrestrial transformations. The "W"-weathering values were systematically taken from the classification schemes in the MetBull and considered in a first approach to W0 for falls.

The two columns describing the degree of alteration / petrologic type need to be explained and are not up-to-date, incomplete, or incorrect. The best starting point for the petrologic classifications is Rubin et al. (2007) GCA 71:2361 (not the Rubin (2007) that is currently cited), while it should also be noted that many of these meteorites are breccias with lithologies altered to different degrees (e.g. Paris is a 2.9 – 2.6). Rubin didn't look at the CI chondrites and calling them 1 doesn't make sense in this column. Also, petrographic classifications have been reported for other meteorites e.g., Aguas Zarcas, Maribo.

The Alexander et al. (2013) classification scheme is based on bulk H and C abundances (not petrography). Again, some data is missing (e.g., Flensburg) and it should be made clear in the caption that a 2.0 in the Rubin scale is not the same degree of alteration as a 2.0 in the Alexander scale. Not to confuse things further, but a note that bulk phyllosilicate abundances have been reported for meteorites such as Moapa Valley and Mukundpura, where no petrologic classification is given in Table S1.

→ The values in the table are the average ones I was compiling at the time of the modelling study and I was using for the models ... as reported in the cited literature. The exact values in the literature and the different existing scales themselves followed many debates over the last decade and generally because these materials are brecciated; these values cover always a wide range as you mentioned in your comments. Bischoff nicely showed in his several papers along his career the brecciation of these materials. For modelling purposes sometimes taken as average/median values of the found ranges.

The trend in model with these calculations is however not affected in little changes in the values.

The caption needs to briefly explain what TI means. This is taken from Quirico et al. (2018), but is really a combination of XRD/mineral observations of Nakamura (2005) and the Raman/IR data of Quirico et al., to estimate peak temperatures for carbonaceous chondrites. The TI meteorites experienced low temperature (<300°C) aqueous alteration and do not record any post-hydration metamorphism (i.e., TII – IV in Quirico et al.). There is published XRD data for many of the samples in Table S1 that is consistent with them being TI (e.g., Howard et al. 2015 GCA 149:206, King et al. 2017 M&PS 52:1197).

→ I added this information in the caption

Fig. S1 caption: Orgueil should be CI1 (if using CM2 for Murchison). Also, typo – green.

→ Done

Fig. S2 caption: The bubble size expresses... ...and the colors correspond... ...One clearly observes... Also, define n.t. in the figure captions.

→ Done

Fig. S3 caption: m/z expanded? ...One clearly observes...

→ Changed to "mass edited H/C" ratios as I Fig.4.

Fig. S8a: either ...an enhanced molecular interaction... or ...enhanced molecular interactions... ...A0106 showed a larger proportion of...

→ Done

Reviewer #2 (Remarks to the Author):

This study highlights the high diversity of Ryugu organic matter using mass spectrometry and NMR spectroscopy. The results are interesting (though redundant with existing literature) and worth publishing in Nature Communications. I made comments, hereafter provided line by line, that I think will help the reader better understand.

→ We thank the reviewer for his constructive comments.

Page 2, line 3 : Please rephrase “since the parent body”.

→ Done

4-7 : Please remove “systematically” and explain why you couldn’t perform ESI(+) on hexane extract and APPI(+) on water extract.

→ Explained in the caption of the figure in SI.

4-32 : What S/N did you use to reduce the 200,000 signals ? What type of information have you possibly lost ?

→ I used S/N higher 3 to extract the data. These contained many signals from the background and all isotopologues. 200.000 was counted without checking overlaps, just summing all signals in the various extracts and ionization modes.
It was important to consider a conservative way to limit any false positive assignments.

4-22 : Table x is Table S1 ?

→ Done

Figure 1 (E) : Please invert SO and S columns.

→ Done

8-3 : “moderate” in the text corresponds to “warm hydrothermalism” in the figure 4 (A) ?

Figure 4 (A) : How exactly did you chose the 2 parameters of the PCA ? The “warm hydrothermalism” annotation corresponds to which CCs ? Those in the upper left corner of the PCA ?

→ We changed to “warm”
The warm hydrothermalism correspond to all green dots in the figure.

9-31 : Do you consider Tarda highly altered only because there is no organomagnesium compounds in it ?

→ Tarda was predicted as well by our model model as highly altered confirming also our earlier published finding

10-16 : Please rephrase the last sentence of the paragraph.

→ done

11-5 : Please update Naraoka paper.

→ Done

Table S2 : Missing figure names ?

→ Done

Figure S1 : Please highlights both C₁₇H₃₅O₃S and C₂₀H₃₁O₅.

→ Done in the caption

Typos :

Page 2, line 7 : shines

3-30 : PAH

4-24 : mass instead of masse

9-25 : various

10-7 : affection

10-15 sulfide

10-21 : Please, leave only one "seen".

10-26 : reflects

19-9 : green

→ Done thank you